# Animals as Something More Than Mere Property: Interweaving Green Criminology and Law

**James Gacek** [1,*] and **Richard Jochelson** [2]

1   Department of Justice Studies, University of Regina, Regina, SK S4S 0A2, Canada
2   Faculty of Law (Robson Hall), University of Manitoba, Winnipeg, MB R3T 2N2, Canada; Richard.Jochelson@umanitoba.ca
*   Correspondence: James.Gacek@uregina.ca

**Abstract:** Our article argues that non-human animals deserve to be treated as something more than property to be abused, exploited, or expended. Such an examination lies at the heart of green criminology and law—an intersection of which we consider more thoroughly. Drawing upon our respective and collective works, we endeavor to engage in a discussion that highlights the significance of green criminology for law and suggests how law can provide opportunities to further green criminological inquiry. How the law is acutely relevant for constituting the animal goes hand in glove with how humanness and animality are embedded deeply in the construction of law and society. We contend that, when paired together, green criminology and law have the potential to reconstitute the animal as something more than mere property within law, shed light on the anthropocentric logics at play within the criminal justice system, and promote positive changes to animal cruelty legislation. Scholarship could benefit greatly from moving into new lines of inquiry that emphasize "more-than-human legalities". Such inquiry has the power to promote the advocacy-oriented scholarship of animal rights and species justice.

**Keywords:** bestiality; companion animals; green criminology; harm; law; police animals

---

## 1. Introduction

Billions of animals[1] continue to be oppressed, exploited and devalued in parts of Western society, and to add to this oppression, "animals have and continue to be perceived as living property in our legal system[s], which [are] conceived by and for human beings" (Verbora 2015, pp. 62–63). While legal protections afforded to animals have improved, what role does green criminology play in this discussion? Can green criminological insight move the legal dial towards substantial progress for species justice? Conversely, can law contribute new perspectives into green criminological scholarship? Is it possible for the pair to interdigitate and thereby advance an interdisciplinary discussion?

Our article intends to answer these questions in the affirmative, and endeavors to engage in a discussion which highlights the significance of green criminology for law[2] and the ways in which law can provide opportunities to further green criminological inquiry. Building on considerable green criminological and legal scholarship, we believe that such an examination of where these disciplines

---

[1]   Sollund (2015b), drawing upon Beirne (2011), takes issue with the term "animals" specifically as the term, per Sollund (2015b, p. 163) "conceals the fact that humans are also animals, and that what is included in the term (by humans) is a large diversity of individuals of thousands of species, rather than one which are thereby contrasted with humans." Despite its problematic nature and a lack of good alternatives, while we will refer to animals in this article, we are referring to non-human animals.

[2]   While beyond the scope of our paper, we recognize that, depending on the system of law, the distinct legal systems active in various international contexts might affect or complicate the analysis of animal welfare laws.

---

intersect will showcase the significance each has for the other. When paired together, green criminology and law can work in tandem to push the bounds of academic interest concerning: (1) injustices towards species other than humans; and (2) activities in society which are harmful yet legal. While the law is acutely relevant for defining the animal, how "humanness" and "animality" are embedded deeply in the construction of law and society is a consideration green criminology brings to the fore (e.g., Gacek 2017). With green criminology, law's anthropocentric orientation becomes clear—a perspective that can be cruel and coercive.

Yet, law, especially in common law systems, is also iterative and reiterative (Jochelson and Gacek 2018); it is inherently responsive to breakdowns in social cohesion and, ultimately, will bend or break to attend to governing cultures. Sometimes this occurs through law reform movements (for example, the anti-abortion movements in Canada and the United States (US)), and other times through progressive judicial reasoning (for example, Canada's anti-hate speech laws emerged despite tremendous pressure on the judicial branch by far right lobbies[3]. Yet, at other times, change can occur when an unattended issue has undergone tremendous social metamorphosis, such that rights based instruments (e.g., constitutions, rights codes), pressure legal actors to push for attendant change (as was the case when Canada's prohibition on prostitution was declared unconstitutional in Canada in *(AG) v Bedford* 2013)[4]. These examples also reveal law's enduring connection to "the social." Whether laws alter, bend, break or inure, there is reflection and refraction of "the social" in its compositions, and it is the tethering of "the social" and law that provides potentialities for progressive (and at times regressive) change.

These tethering points provide ample opportunity for animal welfare and cruelty legislation, and perhaps more progressive instruments of animal entitlements, to present opportunities for green criminological perspectives to inform the reconstitution and reform of these, at times, antediluvian strictures of law. As Jochelson notes (Jochelson 2014, p. 250):

> The relationship of law and the social is complex. Society informs law and law informs society. Neither do so in linear ways. Yet these interactions reveal that law is inherently tied to the social. This ligature makes law inherently transitory and contingent even as it binds the citizen. This dynamism, is described by Golder and Fitzpatrick as law's "alterity;" law has an ability to be other than what it was (i.e., to change), while still retaining its capacity of coercion and manipulation. (Golder and Fitzpatrick 2009)

With the pliability of law, green criminology gains a bountiful array of possibilities within the legal case itself to further interrogate the environmental and animal-related harms permeating society at large. Indeed, in past studies, we have noted that judicial prose, especially in multi-authored legal decisions, can provide alternative constructions of the animal (Gacek and Jochelson 2017a, pp. 237, 240):

> The judicial decision in particular is intriguing legal media because the judiciary, as arbiters of legal issues before a court, interpret law. Interpretation, though, sometimes leads to reconstitution of law, and what was once innocent behaviour can be reconstituted as, for example, criminal through the adjudicative exercise (and vice versa). Certainly, the judicial decision is capable of shifting legal precedents to align with modern contexts of the law, but it is also capable of toeing a conservative statutory interpretation upholding the original Parliamentary intent that animated a statute when it was drafted . . . However, in effect, the judiciary also has the ability to (critically) shift legal discourse to (re)position power relations and social inequalities between humans and non-human beings.

Animal welfare is a rising topic of public interest in many societies, particularly in the West; as Nurse (2016, p. 174) indicates, "Governments have increasingly sought to adopt minimum

---

[3]　R. v. Keegstra. 1990. 3 SCR 697.
[4]　(AG) v Bedford 2013 SCC 72, [2013] 3 SCR 110).

animal welfare standards by enshrining these in legislation." Legal systems sometimes struggle with conceptions of violence and cruelty towards animals; however, they often view animal protection solely as an environmental or welfare issue (Nurse 2013) and do not recognize how the law, in general, and criminal law, in particular, should be more reflective of practices by humans and the animals they own (Gacek and Jochelson 2017a, 2017b; Jochelson and Gacek 2018). The fact that animals are generally regarded as property reflects the anthropocentric nature of law; as Nurse (2016, p. 175) puts it, "animals are generally protected only to the extent that their welfare coincides with human interests."

Following Sollund (2015a, p. 8) and Brisman (2017, p. 313), we see little purchase in "'[policing]' the borders of the field" in order to delineate what research areas should be included or excluded in this discussion, nor is there benefit in determining what research topics should be permitted and what should not. We endorse a more capacious conception which sees value in both green criminology and law as disciplines which may, at times, overlap and sync harmoniously. Such overlaps have potentially fruitful consequences, not only in taking a modest step forward in interdisciplinarity, but in the development of workable solutions and outcomes produced through a green criminological and legal intersection.

## 2. The Significance of Green Criminology for Law

The emergence and development of a green criminological perspective within the last thirty years reflects how little attention criminology has paid previously to environmental and animal-related issues. While the umbrella concept of "green criminology" does not necessitate specific adherence to a set of theories or methods, green criminology provides an apt and warranted discussion of injustice within and throughout our anthropocentric social worlds (Gacek 2017; Hall 2014; Ruggiero and South 2010). White argues that "there is no green criminological *theory* as such" (2013: 22 (emphasis in original)). Perhaps there is no need for one; those who work within green criminology define it in ways that best suit their own interpretations of how green criminology should be applied both in theory and praxis. Indeed, South (2014, p. 8) contends that while there is currently no universal consensus about terminology or applicability within the criminological field, criminologists who critique the environmental and animal-related harms existing in society most frequently define "green criminology" as the study of harm, crime and injustice "related to the environment and to species other than humans."

Yet, the significance of a legalistic approach to green criminology cannot go unnoticed because the law and its analysis has been a central component of green criminological research (see, e.g., Beirne 2009; Benton 1998; Goyes and Sollund 2016; Sollund 2013, 2015a, 2015b, 2017a, 2017b; and White 2013). In particular, one can ascribe the connection between the law and green criminology to a greater focus of green cultural criminology (Brisman et al. 2014; see also McClanahan 2014; Brisman and South 2013, 2014) because legal forces are cultural forces and a legal analysis functions like a cultural one to reveal important tendencies in the construction of animality and justice. Therefore, green cultural criminology highlights the significance of cultural conditions and forces that shape how we perceive and think about environmental harm and harm towards animals. As we have also noted above, law reflects, refracts, absorbs, and feeds into and back from the social world. Law is susceptible to societal shifts and, thus, cultural shifts, and the law functions in agonistic, parasitic, and symbiotic ways with the social order (and vice versa). In other words, how we think about animals has a connection with prospective legal change because culture and law are interwoven, and law is both iterative and reiterative.

Green criminology's blossoming as a key area of debate, moreover, was supplemented by legal scholarship turning towards environmental harms and the impact of such harms on ecosystems and neighboring communities (Hall 2014). In particular, green criminology has considered the rights of victims and the options for redress under the law (e.g., Hall 2011, 2013, 2014, 2015, 2016; Jarrell and Ozym 2012). Indeed, Hall's (2014, p. 97) insight has been significant in not only suggesting how law and legal analysis fit within green criminological inquiry, but in attempting to delineate why and how a legal perspective "is not only a desirable aspect of green criminology but a vital one" for continued

(and further) interdisciplinarity. Hall (2014, p. 103) draws on environmental risk assessments and regulations in developed nations to indicate that "increasingly the general trend in most jurisdictions is for public authorities to view environmentally destructive activities as an exercise in the management of risk." In doing so, Hall engages in a wider green criminological discussion which encompasses both legalistic and non-legal responses in the study and scrutiny of risk management and regulation. As Hall (2014, p. 106) contends, "many of the issues surrounding environmental risk and regulation are fundamentally *legalistic* questions even when 'official' . . . legal censure is absent" (emphasis in original). We concur with Hall that green criminology must incorporate legal analysis as a significant feature of its overall project "if green criminology is to succeed in providing workable solutions to the vast array of problems within the 'environmental' sphere" (Hall 2014, p. 106).

We recognize that there is much complexity in addressing environmental degradation through law, especially criminal law, which may call into question why a further examination of crime, criminals, and criminality within the environmental sphere is warranted. As White (2009, p. 483) puts forth, environmental crime "is studied for a reason; namely, we need to understand the genesis and dynamics of such crime so that we can adequately respond to it." We echo White's sentiment that more work "needs to be done to understand the nature and scope of environmental harm" (2009, p. 483), and we believe that our focus on harms to animals which are (still) legal provides not only necessary attention to a contentious aspect of law but supplements a greater consideration for "green" issues at large. Concern for animals and the ecosystems that comprise their habitats are inherently linked to environmental concerns. Further law reform, whether packaged as "green reform," animal welfare legislation, or even as something more radical, such as a "green constitution,"—the latter of which grants rights to nature—has the potential to propagate societal understandings of human–animal relations and galvanize the discussion about appropriate and just treatment of animals in Western, liberal democracies. Informed by green criminological insight, law could recognize the socio-political and anthropocentric machinations of the criminal justice system and reconsider, for example, what safeguards must be implemented to ensure animals conscripted for police work are secured from intrusive or harmful practices (Gacek 2017), or how morality and animal sentience are, or should be, implicated in animal welfare and anti-cruelty legislation (Gacek and Jochelson 2017a, 2017b; Jochelson and Gacek 2018).

Green criminology's attention to harms against and injustices towards animals has the potential to recognize the inherent sociality of the legal case. Such recognition can benefit the judiciary not only in reading and adjudicating the legal case before them, but in reconsidering legal definitions, discourses and harms in the context of shifting socio-political landscapes. As we demonstrate below, our present relationships with animals are a social construction, not a historical or natural constant (Sorenson 2010; Gacek 2017; Van Uhm 2018). Re-examining and re-evaluating the harms caused to animals is significant for exposing anthropocentric logics at play in the criminal justice system.

*Case Study: Police Animals and Species Justice*

As Brisman (2014a, p. 25) contends, more research needs to be undertaken "to understand the ways in which environmental crime and harm are *constructed by* and *represented in* the media and the ways those constructions and representations affect how we ascribe meaning to the environment, to nature, and to harms and crimes thereto" (emphasis in original). Examples have included Brisman and South's (2013, 2014) examination of how the media portrays real or imagined environmental harms and disasters, and the mediated and political dynamics surrounding these presentations in the news. Similarly, Kohm and Greenhill (2013) have explored "popular" issues within the media, drawing on media depictions of environmental harm, issues of place and space, and oppression relations between humans and animals to characterize a greater concern for the interconnections between the nature of harm and social and physical environments in film and television. While we contend that cultural representations and shifts are interwoven with law and legal change—and thus that the study of

culture has repercussions for the study of law—we note that much of the aforementioned green cultural criminology leaves it to the reader to consider the implications for law.

Just as media and cultural constructions and representations of environmental harm and crime are significant and worthy of study, so, too, are the legal constructions and representations that are caught up and bound within understandings of harm and animality. As indicated above, by drawing together green criminology's consideration of both culture and law, we begin to recognize that as a cultural product itself, law also functions to construct animality and the harms associated by and through such a construction. For example, Wall (2014, 2016) examines the symbolism intertwined in the use of the police dog as a technology of suppression against Black people. As Wall (2016, p. 861) suggests, for many minority communities—both in periods of civil unrest and protest and in daily life—"the animalization of police power prove[s] symptomatic of a much-longer history of [W]hite organized violence against the [B]lack community." Wall contends that the police dog exists as a tool of racial and legal terror enacted through police power. His research reveals the ways in which police animals are used to sustain the liberal, capitalist order, suggesting that the police dog is "not only a metaphor for sovereign monstrosity" (Wall 2014, p. 4) but a "key symbolic figure of racist state violence that place[s] historical subjugation in conversation with the present" (Wall 2016, p. 861). Especially in the era of social media, such images of police dogs attacking Black people contribute to the construction of the dog as both "flesh and symbol" of the police's ability to "take a bite out of crime" and the law's ability to "make and unmake personhood in fundamentally racialized ways" (Wall 2016, p. 862). Taken together, such constructions of police animality "conjur[e] up the apparitions of historical violence to hauntingly appear in the space of the present" (Wall 2016, p. 862) and perpetuate harms towards marginalized communities.

In a similar vein, understanding the representations and meanings of crime and harm bound within police animality has been problematized from a green criminological perspective (Gacek 2017). For example, Gacek questions the anthropocentric logics at play in police operations involving police animals and highlights the need to focus on "species justice" (see also Beirne 2011; White 2011; Wyatt 2011). Drawing on green criminology and critical security studies, Gacek argues that non-human animals should bear rights like their human counterparts, and that from a species justice perspective, police eagles (and other police animals) have the inherent right not to suffer abuse from humans through this type of police work. Through a qualitative media/thematic analysis, Gacek examined twenty media reports about the Dutch "Flying Squad"—a convocation of eagles purchased by Dutch national police. Gacek's study was inspired by Braverman's (2013, 2015) work concerning the "police animal" categorizations, in which Braverman charts an insightful and complex understanding of the ongoing relationship between nature and technology existing within the form of the police dog. In a similar vein to Braverman, Gacek (2017) argues that rather than placing the eagle in "*either* the 'nature' box *or* that of 'technology'" eagles and dogs used for police work exist in both; the "bio" component refers to the animal's "aliveness," which does nothing to negate its technological use to for humans (Braverman 2013, p. 7 (emphasis in original)). Essentially, the police animal constitutes a form of "bio-technology" for the state that is coproduced between the animal and the handler. In effect, police animals are "humanly crafted means to [serve] humanly formed ends and desires" (Braverman 2013, p. 7) which allows humans to subjugate their police animal counterparts to further securitize human society, often at the expense of the animals involved (Gacek 2017, p. 2).

Gacek's findings suggest that within the media reports, the representation of the police eagles as the "most effective countermeasure" to combat drones in Dutch skies perpetuates an anthropocentric logic which can be accepted easily by governments and police agencies to justify their efforts to securitize the skies. Such logic is exacerbated within the hyper-exaggeration of the police eagles' "animal instincts"—a view that constructs the eagle as a set of desirable skills and characteristics to be used by humans rather than a being with potential sentience (Gacek 2017, p. 11). In Braverman's (2013, p. 27) examination of sniffer and detection police dogs, the heightened olfactory sense of the dog was constructed by police agencies as an "advancing technology" that could be used to the advantage of

police officers engaging in search and seizure operations. Similarly, Dutch police and "Guard From Above," a security firm specializing in eagle training, constructed the "increased visual acuity" of the eagles as advantageous to police operations in Dutch skies (Gacek 2017, p. 8). Dutch police and "Guard From Above" coupled this notion (of the eagle's "increased visual acuity") with the eagle's "natural suspicion of drones" without providing any substantiated empirical evidence to support such a claim (Gacek 2017, p. 8). Finally, the minimalization of news coverage concerning the rights of the eagles, as well as the lack of consideration for the eagles' welfare while in captivity, demonstrates the extent of neglect these organizations have in safeguarding the police eagles.

Gacek examines Dutch animal cruelty laws and questions whether such laws are appropriate for protecting the eagles from unreasonable harm, injury, and stress. In the Netherlands, *the Animals Act* (Government of the Netherlands 2011) provides for the protection of animals in captivity, specifically in Article 2.1 ("Animal Cruelty") and Article 2.2 ("Keeping Animals"). This legislation prohibits the infliction of pain or injury on an animal, or the damaging of an animal's health and welfare without reasonable purpose or more than what is reasonable for such purpose. According to *the Animals Act*, any violation of these animal cruelty provisions is a criminalizable offense punishable by fines or imprisonment of up to six months as set forth in Article 8.12 ("Penalties") (Gacek 2017). The legislation, however, offers another indication of uncertainty for police animals—much like the media reports. Gacek found that while the protection outlined in *the Animals Act* applies to domesticated animals, it remains unclear whether such protections are extended to wild or stray animals. Moreover, Gacek underscored how the legislation does not find cruelty problematic when considering police animals that are not "possessed" in the same manner as agriculture or companion animals.

In sum, species justice is—or should be—a contemporary issue of law and justice, and a species justice approach can assist criminology and law in understanding how each discipline has understood and constituted animals (Beirne 2007; Nurse 2011, 2016; Spencer and Fitzgerald 2015). Legal constructions and representations of police animality are bound within cultural conditions and forces that inure with society; yet, as we can see, sometimes the anthropocentric logics embedded within these constructions of police animality discount, evade, or suppress the harms such animals face in police work. Whether the harms to animals entail "one-on-one harm, institutionalized harm" or harm "arising from human actions that affect climates and environments on a global scale" (White 2011, p. 23), the inclusion of eagles into the "police animal" category perpetuates anthropocentric logics within the law and generates a new bio-technological mode of policing which reifies a worrisome norm for criminal justice systems in the West: perceived human security (including the mere fear of human harm) is worth the subordination of non-human animals. By examining law more closely, there may be more fruitful endeavors available for green criminology to achieve "just" reforms in animal welfare and protection.

## 3. The Significance of Law to Green Criminology

As described above, law is significant to the construction and placement of animals in society and through judicial decisions, we see how potential discursive shifts in such issues may be reflective or refractive of societal perceptions once the legal text is generated through case adjudication. As Cotterell (2006, p. 25) suggests, law "is an aspect or field of social experience, not some mysterious work working upon it." While law can reflect how citizens view justice, it can also be a coercive regulator of behavior; it can be iterative and reiterative. Social interpretations of law—cognizant of social changes and modern contexts—can feed back into the formal legal system, subsequently leading to delegated social governance in communities, as well as in civil society more broadly (Jochelson et al. 2017b, p. 107). Law, then, is not a truth per se, but a "mobile and contingent" feature of the social ties that bind (Golder and Fitzpatrick 2009, p. 125). As we have argued elsewhere:

> Simply stated, the judiciary has the power to alter legal conceptions through case adjudication. However, in effect, the judiciary also has the ability to (critically) shift legal discourse to (re)position power relations and social inequalities between humans and non-human beings.

> In effect, we argue that "legal language is a socially constructed institution in its own right" (Stygall 1994, p. 4). This can be justified through the underpinned logics and judicial articulations within legal text. (Gacek and Jochelson 2017a, p. 240)

What we find troubling, however, is that while human beings, in Western democracies, typically possess legal rights, rights to ensure that our fundamental interests (such as our interest in life, liberty, and security of the person) cannot be overridden—except in limited circumstances and on a principled basis (depending on the relevant constitutional instrument at play)—the same cannot be said for animals (Gacek and Jochelson 2017b, pp. 336–37). Animals do not possess anything approaching the guaranteed rights and protections of persons outlined in constitutional documents such as in Canada or the United States—a view that concerns many scholars who advocate for shifts in such legislation (Jochelson and Gacek 2018; Sankoff 2012; Verbora 2015; see also Brisman 2014b for a review and consideration of interdigitating environmental rights with human rights).

Propelled by science and ethics, public interest in animal issues is mounting, and in legal scholarship, there has been, within the past decade, an increasing amount of debate for animal welfare reforms in both Canada and the United States. The pitfalls of animal welfare legislation in the West have become more prominent than ever and there is rising pressure for law reform to ensure that animal welfare be reflective of contemporary insights and values. Such reform has been advocated by animal rights activists and scholars (Bisgould 2014; Bisgould and Sankoff 2015; Sorenson 2010; Sykes 2015), indicating that accumulated scientific knowledge has demonstrated that animals are more than property: they are "beings with emotions, consciousness, and sentience; yet legal regulations often administer animals as mechanistic property, to be utilized by human beings" (Gacek and Jochelson 2017b, p. 336; see also Deckha 2012; Sankoff et al. 2015). In fact, animals in Canada are arguably less safe than in the Ukraine or in the Philippines, both of which have stronger legislation in place to protect them (Verbora 2015). Regardless of which side of the border one calls home, it is clear that animals in Canada and the United States need better legal protection from deliberate acts of cruelty or negligence. It is also clear that broad social movements have not coalesced successfully around the contention that animals in North America ought to be rights-bearing subjects. Certainly, some constitutional protections for animals are observable internationally; for example, in Switzerland, the law overtly protects the dignity of animals and is constitutionalized. This protection, however, situates animal proto-rights as inherently subordinate to human rights: the protections serve only to counterbalance human interests and do not extend to immanent standalone guarantees for animals (Bolliger 2016; Jochelson and Gacek 2018).

Interestingly, legal cases can be significant sources of information for green criminology because the cases contain analysis, argument, context, history, precedent, and reasoning. Indeed, judicial opinions and orders, especially at the highest levels of the legal system, are not pulled from the ether; they draw upon previous court decisions and social changes that preceded the case at hand. According to Berlant (2007, p. 663), "the case represents a problem-event that has animated some kind of judgement," which may speak to greater societal concerns at large. Examining specific cases also enables the researcher to show how disparate expert knowledges can fold space and time to produce an "event" in the present (Berlant 2007; see also Ettlinger 2011). Doing this allows for analysis to include both the social processes beginning outside of the law which have become "juridified,"[5] as well as accounting for the ways that law structures decisions that govern social outcomes (Hunt 1997; Jochelson et al. 2017b). As Jochelson et al. (2014, p. 19) point out, while it is important for courts at the highest levels to understand past law to articulate, constitute, and/or amend legal tests, "[i]t is just as

---

[5] In other words, and as we have suggested elsewhere (Gacek and Jochelson 2020, p. 12), studying the logics underpinning legal texts ripened with "judiciomentalities" (i.e., legal expressions that imbed social constructions of history, politics, precedential strictures, constitutionalism, and personal/political judgement) allows us to consider legal texts themselves as representative of a type of technology that delivers and rationalizes the governmental effects of law separate and apart from the law that itself created.

important to place these constructions in a socio-political place by analyzing the social conditions that inured in the relevant era. By necessity this involves a contextual and careful line-by-line reading of the decision to be examined." Therefore, the case as a pedagogical tool (Berlant 2007; Brisman 2010) can be instrumental in cultivating critical discourses which counter anthropocentrism in the law and the criminal justice system at large.

Like other decisions that have extended rights to humans where it was previously thought the common law stunted progress and change (Jochelson and Kramar 2005), perhaps it is time to reconsider the legal position of animals. We follow Nurse's (2016, p. 185) assertion that "the public benefits of animal welfare must be weighed in the context of prevailing social conditions" and, as we have indicated in previous work (Gacek and Jochelson 2017a, 2017b; Jochelson and Gacek 2018), social considerations for animal welfare are shifting—albeit incrementally—in this spirit. Perhaps it is time for the courts to interpret laws that implicate animals in light of potential sentience and constitute the animal as a being that is worthy of, at minimum, modest protections and immanent worth—a discussion to which we now turn.

*Case Study: Constituting the Canine in Case Law*

Two recent cases, *R. v. D.L.W.* (2016), before the Supreme Court of Canada (SCC), and *State v. Newcomb*, before the Oregon Supreme Court, considered what it means to be an animal in situations of bestiality and animal welfare investigations specifically. We analyzed the trial, appellate and SCC decisions of *D.L.W.* and undertook a similar analysis in *Newcomb*, mining the legal text for judicial reasoning and rhetoric pertaining to the interpretation of the legal terms "bestiality," in the former, and "property," in the latter. Our research and analysis sought to examine the logic of the courts in the *D.L.W.* and *Newcomb* decisions, and by analyzing each case in turn, understand the construction of the animal in these cases.

In 2016, the SCC heard an appeal from a decision from the British Columbia Court of Appeal, which provided a narrow interpretation of the offense of "bestiality" (*R. v. D.L.W.* 2016, p. 403)[6]. In *D.L.W.*, the appellant was charged with a total of fourteen sexual offenses involving his two stepchildren. The appellant was then found guilty on thirteen counts by the trial judge in the Superior Court of British Columbia, including one count of bestiality. The bestiality charge emerged from an incident, which was non-penetrative in nature, caused by the accused and that involved the family dog and a stepdaughter.

Prior to the recent amendments of December 2019, which broadened the definition of bestiality to include sexual touching of any sort with an animal, the Criminal Code{ XE "Criminal Code: section 160" } of Canada prohibited "bestiality" in Section 160 (Criminal Code *R.S.C.* 1985, c. C-46). The word "bestiality" was judicially interpreted in an extremely narrow manner as criminalizing penetrative offences involving genitals. The Code continues to delineate three separate offences. Section 160(1) deals with the basic offence of bestiality by the accused:

(1) Every person who commits bestiality is guilty of an indictable offence and liable to imprisonment for a term not exceeding ten years or is guilty of an offence punishable on summary conviction.

Section 160(2) criminalized situations where the accused has compelled another person to commit bestiality:

(2) Every person who compels another to commit bestiality is guilty of an indictable offence and liable to imprisonment for a term not exceeding ten years or is guilty of an offence punishable on summary conviction.

Finally, section 160(3) criminalized situations where an accused either commits bestiality in the presence of someone under the age of 16, or who causes a person under the age of 16 to commit

---

[6]    *R. v. D.L.W.* 2016. SCC 22.

bestiality themselves. It bears mention that the punishment for this offense is greater than that of the other two; this offence has a mandatory minimum sentence, as well as a maximum sentence which is four years greater than the maximum carried by the other two offenses:

(3)   Despite subsection (1), every person who commits bestiality in the presence of a person under the age of 16 years, or who incites a person under the age of 16 years to commit bestiality,

    a.   is guilty of an indictable offence and is liable to imprisonment for a term of not more than 14 years and to a minimum punishment of imprisonment for a term of one year; or

    b.   is guilty of an offence punishable on summary conviction and is liable to imprisonment for a term of not more than two years less a day and to a minimum punishment of imprisonment for a term of six months.

Other sections of the Criminal Code of Canada contemplate offenses against animals, but it is less clear whether bestiality can amount to the degree of harm required to find an offense under these sections. For example:

445.1 (1) Everyone commits an offence who

a.   willfully causes or, being the owner, willfully permits to be caused unnecessary pain, suffering, or injury to an animal{ XE "animal" } or a bird;

The punishment under this section is set out in section 445.1(2):

(2) Everyone who commits an offence under subsection (1) is guilty of

a.   an indictable offence and liable to imprisonment for a term of not more than five years; or

b.   an offence punishable on summary conviction and liable to a fine not exceeding ten thousand dollars or to imprisonment for a term of not more than eighteen months or to both.

The difficulty with applying Section 445 to situations of bestiality is that these provisions require proof of harm to the animal{ XE "animal" }, which could be very difficult to establish without the presence of an obvious injury or expert examination of the animal close in time to the alleged act. Arguably, this means that Section 445 could capture an even narrower range of sexual conduct than bestiality under Section 160.

The term, "bestiality," tends to be used to refer to sexual relations between humans and animals (see, e.g., Beirne 1997). Beirne (1997, p. 320) suggests that usually, "in law, [bestiality] refers to sexual intercourse when a human penis or digit enters the vagina, anus or cloaca of the animal. However, it often also entails any form of oral-genital contact, including those between women and animals, and even, in psychiatry, fantasies about sex with animals."[7] Yet, prior to the recent Parliamentary amendments, the SCC, with a majority of six-to-one, determined that "carnal knowledge" (i.e., penetration) was an integral factor in the definition of bestiality (*R. v. D.L.W.* 2016, p. 402). In its decision, the majority noted that the scope of both bestiality and criminal liability at large must be determined by the Canadian Parliament (*R. v. D.L.W.* 2016, p. 3). In the majority's opinion, judges "are not to change the elements of crimes in ways that seem to them to better suit the circumstances of a particular case" (*R. v. D.L.W.* 2016, p. 3). Supreme Court Justice Thomas Albert Cromwell, in writing for the SCC majority, noted that "the old case law is not abundant, but what there is supports the view that penetration was an essential element of the offence" (*R. v. D.L.W.* 2016, p. 33) and "whatever [bestiality] was called [throughout history], the offence required penetration" (*R. v. D.L.W.* 2016, p. 24).

According to the SCC majority, the early legal history of bestiality in Canada was subsumed under the offenses of "sodomy" or "buggery" and that penetration was certainly one of the offense's essential elements (*R. v. D.L.W.* 2016, p. 50). In addition, the SCC majority noted that despite comprehensive

---

[7]   For a detailed consideration of the philosophical issues raised in the language and spirit of bestiality laws, see Beirne (1997).

revisions and amendments of sexual offenses throughout Canadian legislative history, the Parliament of Canada (the "Parliament") never sought to change the common law definition of bestiality (*R. v. D.L.W.* 2016, p. 52). In the eyes of the SCC majority, this demonstrated a clear indication that Parliament's intention to retain the term was "well-established" (*R. v. D.L.W.* 2016, p. 19) and the definition of "bestiality," itself, had a "well-understood legal meaning" (*R. v. D.L.W.* 2016, p. 18).

Supreme Court Justice Rosalie Abella dissented:

> [D.L.W.] is about statutory interpretation, a fertile field where deductions are routinely harvested from words and intentions planted by legislatures. But when, as in this case, the roots are old, deep and gnarled, it is much harder to know what was planted.

> We are dealing here with an offence that is centuries old. I have a great difficulty accepting that in its modernizing amendments to the Criminal Code, Parliament forgot to bring the offence out of the Middle Ages. There is no doubt that a good case can be made, as the majority has carefully done, that retaining penetration as an element of bestiality was in fact Parliament's intention.

> But I think a good case can also be made that...Parliament intended, or at the very least assumed, that penetration was irrelevant. This, in my respectful view, is a deduction easily justified by the language, history, and evolving social landscape of the bestiality provision. (*R. v. D.L.W.* 2016, pp. 125–27 (emphasis added))

In sum, Justice Abella argued that imposing the penetrative component of "buggery" on legal definitions of bestiality would leave "as perfectly legal" all sexually exploitative acts with animals that do not involve carnal knowledge (*R. v. D.L.W.* 2016, p. 142).

The case of *D.L.W.* is illustrative of a dissent that considers the harms to animals by humans, the integrity of the animal violated, and the cruelty to animals who are vulnerable beings. The majority may have seen these concerns as ancillary, given that the accused was sentenced and punished for the attendant crimes of sexual assault of human youths that occurred together with the bestiality. This anthropocentrism, though, allowed for the troubling conclusion that the sexual touching of an animal outside of coitus was not bestiality—a perspective that remained the prevailing state of the law in Canada until the December 2019 amendments.

Significantly, Justice Abella advanced an argument that reflects growing concern for human–animal relationships (*R. v. D.L.W.* 2016, pp. 140–42). Rather than expand the scope of criminal responsibility (and such power rests not with the judiciary but with Parliament), Justice Abella, seeing an inherent exploitation of animals in bestial acts (based on modern understandings of consent and dignity of all beings) sought to acknowledge the societal concern for animal welfare. Touching animals for sexual gratification is an illegitimate activity in society, and while the current bestiality provision within the Criminal Code of Canada does not provide for the redress for animals who have suffered harm, it is possible that the provision can be changed. Reconsidering bestiality through a green criminological lens allows the law to recognize that bestiality uses vulnerable sentient beings for exploitive purposes and creates needless risks of harm by virtue of the wide range of sexual activities involved and associated with this offense. Proving bestiality occurred, however, is more difficult than simply demonstrating physiological harm to animals; the very nature of bestiality suggests that the act between the human and subjected animal will almost inevitably and typically occur in private (Gacek and Jochelson 2017b). This indicates that only in rare instances will the examination of the animal near the time of offending be possible, which makes it more difficult for police and prosecutors to prove that offenses have, indeed, been committed.[8] Moreover, Beirne (1997, p. 324) contends that while researchers

---

[8]     For a further discussion of the legal-technical nature of environmental crimes and the issues police and prosecutors face, see du Rées (2001).

have examined the physiological consequences of bestiality for humans . . . they pay no such attention to the internal bleeding, ruptured anal passages, the bruised vaginas and the battered cloaca of animals, let alone to animals' physiological and emotional trauma. Such neglect of animal suffering mirrors the broader problem that, even when commentators admit the discursive relevance of animal abuse to the understanding of human societies, they do not perceive it, either theoretically or practically, as an object of study in its own right.

The majority in *D.L.W.* considered legislative intent, seeing the crime of "bestiality" as limited to carnal knowledge (*R. v. D.L.W.* 2016, p. 122). Irrelevant to their calculus were advances in the understanding of consent and emerging social mores about animal sentience. As a result, the majority ignored understandings of the animal as a being, rather than as chattel, and focused only on the moral damage to persons or society at large as a result of immoral sexual behaviors. In effect, for the majority, the animal was merely the circumstance where, or site in which, the criminal act took place. The animal was a mere part of the actus reus of the crime and no harm occurred to a victim.

To be sure, judicial decisions are not immune to shifts in societal perceptions. In *Newcomb*, the Oregon Supreme Court (the "Court") reviewed a case in which the defendant accused the State of Oregon of violating her constitutional rights by taking a blood sample of her dog, Juno, without a warrant to do so. Ultimately, the Court held that the defendant did not have a protected privacy interest in the dog's blood and, therefore, the state did not violate the defendant's constitutional rights. Article I, Section 9, of the Oregon Constitution provides, in part: "No law shall violate the right of the people to be secure in their persons, houses, papers, and effects, against unreasonable search, or seizure . . . " (This language parallels that of the Fourth Amendment to the United States Constitution.) The provision applies only when government officials engage in conduct that amounts to a search or a seizure. A seizure would only have occurred if, through State action, there was a significant interference with the owner's ownership interests in the dog or its fluids. Newcomb argued that under Oregon Revised Statute (ORS) 609.020, which provides that "[d]ogs are . . . personal property," a dog is the same as any other item of property that can be owned or possessed lawfully.

The Court, however, disagreed, noting that an animal raised a different kind of search and seizure issue because the animal is "not an inanimate object or other insentient physical item of some kind" (*State v. Newcomb* 2016, p. 439)[9]. Indeed, the Court explained that an overarching theme reflected in the statutes governing animal mistreatment and neglect under Oregon law is the recognition that some animals are sentient beings capable of experiencing pain, stress, and fear, and what mattered specifically in regards to the case at hand was whether Oregon law prohibits humans from treating their animals in the same ways in which they can (legally) treat other forms of property (*State v. Newcomb* 2016, p. 441). The Court stated that there was probable cause by the officer to believe that Juno required medical attention; the officer could act not only to preserve the evidence of animal neglect but to render aid to a near emaciated canine (*State v. Newcomb* 2016, p. 442).

The Court concluded that Newcomb had no protected privacy interest in Juno's blood, and thus there was no violation of the law by the medical procedures performed by Oregon Humane Society veterinarian, Dr. Hedge, in examining the dog (*State v. Newcomb* 2016, p. 442). Specifically, the Court held that there was probable cause to believe that an animal's welfare was jeopardized by way of malnourishment, and that the drawing and testing of the dog's blood would assist in both diagnosing and treating the dog (*State v. Newcomb* 2016, p. 442).

The *Newcomb* Court reasoned that "Juno is not analogous to, and should not be analyzed as though he were, an opaque inanimate container in which inanimate property or effects were being stored or concealed" (*State v. Newcomb* 2016, p. 442). According to the Court, the "contents" extracted from Juno were, in fact, "more dog" and "the chemical composition of Juno's blood was a product of physiological processes that go on inside of Juno and not 'information' [Newcomb] placed in the

---

9    State v. Newcomb. 2016. 375 P.3d 434 (Or. 2016).

dog for safekeeping or to conceal from public view" (*State v. Newcomb* 2016, pp. 442–43). While the Court was mindful that a dog is considered personal property under Oregon law, which grants animal owners dominion and control over their animals, the Court contended that, simultaneously, Oregon law limits ownership and possessory rights in ways that cannot be equated with other inanimate property (*State v. Newcomb* 2016, p. 443). These "reflections of legal and social norms" ensure that live animals receive basic minimum care and veterinary treatment, and that an animal owner "simply has no cognizable right, in the name of her privacy, to countermand that obligation" to their animal (*State v. Newcomb* 2016, p. 443).

In sum, the *Newcomb* Court indicated that, when assessing the constitutionality of an animal owner's protected privacy interests, such interests of privacy and possession must be contextualized with the "evolving landscape" of social and behavioral norms; such a conceptualization allows for the possibility of acknowledgement of the sentient properties of some animals, and could potentially provide support for the reconfiguration of humans' relations with their non-human counterparts (*State v. Newcomb* 2016, p. 444). While the dog, Juno, was evidence that a crime took place, the Court held that the animal simultaneously occupies both the status of property and that of a quasi-rights bearing subject (*State v. Newcomb* 2016, p. 440). The animal is deserving of health and wellness, and this outweighs the privacy interest of humans (*State v. Newcomb* 2016, p. 441).

In addition, the Court was willing to interpret the Oregon Constitution in light of the evolving social and legislative landscape of animal welfarism (*State v. Newcomb* 2016, p. 444). The result is that the decision recognizes that some animals are sentient beings and that a duty of protection is required for those animals, which can compete with and, on some occasions, override a human's right to privacy (*State v. Newcomb* 2016, pp. 441–43). This is a powerful finding from a state Supreme Court, because the decision seems to build animal proto-rights into a balancing calculus which can mediate or limit a human's right to be free from unreasonable search and seizure at the hands of the state. The human need for a reasonable expectation of privacy may then cede or be qualified by the animal right to life which, in this case, included the right to be free from malnourishment. Reading the decision using these logics demonstrates a potential progressive and radical outcome and demonstrates the iterative and reiterative nature of law in response to the developments of the social world. Green criminology could benefit from fostering these social ligatures and finding ways to make persuasive voices of its philosophies and tenets, such as they are, heard in the legal case. Whether through law reform, intervention in important appellate cases, or more organization in general at a grassroots level, driving the social and directing law reform are twin strategies that complement each other, and which can reflect and refract legal change that calls for dramatic and progressive evolution in the direction of more recognition of animal rights.

Examining property and its attendant crimes and civil wrongs through a green criminological lens highlights that consumption, ownership and care and control are, for the average citizen of common law nations, post-feudal concepts. There is nothing innate or immanent in these conceptualizations of chattel. Nor should conceptions of property be fixed. The use of animals by humans for agriculture, food and, indeed, companionship, is in many respects ancient and accepted. Yet, we live in an era that acknowledges that a living being recognized as property under the law may be entitled to the absence of cruelty—hardly a revolutionary measure, we must admit. However, for now, we can point to *Newcomb* as reflective of the evolving social landscapes of animal wellbeing and the necessity of animal owners to take on a duty of protection towards their non-human counterparts.

In examining these two cases from two different jurisdictions, we see different perspectives on the animal in the socio-legal landscape. In *D.L.W.*, the liberty of the human was given paramount effect over the incursion of sexual abuse on the sentient animal. In *Newcomb*, the Oregon Supreme Court saw the right to privacy in sentient animals as ceding to the entitlement of the sentient animal to live and be safe in the context of animal welfare investigations. In effect, however, the approaches represent different sides of the same coin, as they both understand animals as property and though the implications of sentience inculcate *Newcomb*'s green criminological and species justice affinities

in a more pronounced fashion. While agriculture, manufacturing, farming, and cattle (together with energy, of course) are the lifeblood of these capitalist economies, the Court in *Newcomb* and the dissent in *D.L.W.* represent degrees of incremental resistance and change—and today's resistance in law can be tomorrow's landmark emancipatory decision. The iterative, reiterative, and malleable nature of law provides possibilities and hope for green criminological perspectives to inform the conversation and, thus, the legal status of nonhuman animals.

We view Oregon's recent jurisprudence as a significant step forward in reconsidering the animal alongside contemporary considerations of animal existence and possible sentience. Strict constructions or narrow interpretations of bestiality and the nature of property increase the potential for animals to suffer under the control of their handlers and owners, however. While we lament the absence of such a reconsideration in *D.L.W.*, we remain hopeful that the legal case and judicial decision can be essential sites for progressive changes in animal welfare and legal reform. Indeed, in late 2019, an amendment to the Act resulted in section 160(7) which now defines bestiality as "any contact, for a sexual purpose, with an animal"—an incremental shift, but one that is more protective of animals.

## 4. Conclusion: "More-Than-Human Legalities" Moving Forward

Our article argues that animals deserve to be treated as something more than property for humans to abuse or exploit. As Benton (1994) contends, those who wish to ascribe rights to animals, including the right to respectful treatment, will eventually be forced to challenge the very existence of animals as private property. Property is a word embodying a particular legal relationship and social construction we have chosen to enforce in society. Time and time again, any meaningful effort to achieve more progressive animal protection "quickly collides with [the animals'] entrenched status as things" (Bisgould 2014, p. 162). This does not necessarily have to be the case; as we have indicated, a meaningful study of green criminology and the law's intersection challenges the anthropocentrism of much law. Judicial decisions, which combine context, history, and precedent, provide a wealth of information for green criminology and interdisciplinary academic inquiry, more broadly. Such information has the potential to enrich civic engagement, discussion, and education, and should be welcomed; a critical reconsideration of the landscape of police work and case adjudication can bring to the forefront questions such as "the ways in which we live our socio-legal lives" and how this might impact our understanding of animals as property within human-animal relations (Jochelson et al. 2017b, p. 115).

To be clear, we are not suggesting here that the marriage between green criminology and law is perfect; much like any marriage, such a coupling will have its moments of coalescence and conflict, of triumphs and trials. We recognize there are times where the intersection of green criminology and law will work in harmony for some researchers and provide tensions for others. Indeed, black letter law analyses have their place, as do complex theoretical interrogations of criminal law. As Jochelson et al. (2017a, p. vi) contend, speaking across disciplines between law and cognate disciplines like criminology "is an ever-present challenge." Nevertheless, "[w]e must never forget that good criminal law practice is informed well by social sciences and humanities. [Likewise], the cognate disciplines would also do well to take doctrinal analysis seriously and to include rigorous legal analyses in their own interpretations" (Jochelson et al. 2017a, pp. vi–vii).

Therefore, we see significance in the green criminological and legal pairing, as this scholarship could benefit greatly from moving into new lines of inquiry that emphasize "more-than-human legalities" (Braverman 2015, p. 1). As evidenced from our respective and collective work, such inquiry has the power to promote the advocacy-oriented scholarship of species justice and animal rights. Interweaving green criminology and law is more than a mere academic exercise; the judicial decision can be a site for animal advocacy to happen. As noted above, law is iterative and reiterative; it is malleable. It is mutually constitutive and connected with the social, and there is power in this connection to assist green criminology in modestly resisting anthropocentrism. Bringing green criminology and law together has the power to shift the legal dial in the direction of justice for animals. Incremental drifts towards progressive conceptions of animal existence is possible, and it is through a green

criminological and legal intersection that we can begin to modestly resist anthropocentrism and inch closer to "more-than-property" legal recognition for animals.

**Author Contributions:** J.G. and R.J. were involved in all aspects of the manuscript. J.G. and R.J. drafted, edited and provided a final review of the manuscript. Both authors contributed to the manuscript. All authors have read and agreed to the published version of the manuscript.

**Funding:** This research received no external funding.

**Acknowledgments:** We kindly thank the guest editors, Bill McClanahan and Avi Brisman, for their invitation to contribute to the special issue. This research received no specific grant from any funding agency in the public, commercial or not-for-profit sectors.

**Conflicts of Interest:** The authors declare no conflict of interest.

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
