# Peer review of "Animals as Something More Than Mere Property: Interweaving Green Criminology and Law"

_socsci, doi:10.3390/socsci9070122_

Round 1

Reviewer 1 Report

The manuscript is chaotically written. The general scope of the paper is understood, but the way how it is presented should be dramatically improved. The introduction it seems more a conclusion than the introductory part of the paper. What green criminology is should be explained in the introduction. The case studies are vaguely described. A paper beginning with the sentence: “Billions of animals continue to be oppressed, exploited and devalued in parts of Western society, and to add to this oppression”… have the risk of losing a very important part of the readers that could be benefited of the overall objective of the paper.

Author Response

We do find ourselves agreeing entirely with Reviewer 2's important suggestion that the paper undergo minor revisions in order to further clarify the distinct legal systems active in various international contexts, and how those complexities and distinctions might affect or complicate the analysis of animal welfare laws. We feel that this revision can be made simply, perhaps in a footnote or a short paragraph inserted at the relevant point in the main text.

Reviewer 2 Report

In the present paper, the Authors point out that the animals, today still, are considered a property, and underline as green criminology and law together could have the potential to reconstitute the animal as something more than mere property within law.

I think the topic is relevant and interesting, and is perfectly in line with the scope of the journal. Unfortunately, the Authors remain rather vague for many aspects. I am missing a clear description and a number of relevant details, for example because the animals are more than a property.

The Authors do cite only few court decisions. Could be interesting to report further specific jurisdictions of several countries in order to carry out a comparison.

Canadian law is different than American law and is different than the EU laws, depending on the system of law. Nowhere do the Authors make this clarification to the readers.

In my opinion, the manuscript could be accepted for publication in Social Sciences reporting the above said integrations.

Author Response

While we certainly appreciate Reviewer 1's request for an expanded definition of green criminology, we do not feel that such a revision is necessary--and, moreover, we feel that such revisions would quickly lead to overwhelming redundancy across the special issue--given the clear definitional context provided by the Editors Introduction to the special issue, as well as those definitions provided within individual papers.